# Cardiopulmonary Rehabilitation in Long-COVID-19 Patients with Persistent Breathlessness and Fatigue: The COVID-Rehab Study

**DOI:** 10.3390/ijerph19074133

**Published:** 2022-03-31

**Authors:** Florent Besnier, Béatrice Bérubé, Jacques Malo, Christine Gagnon, Catherine-Alexandra Grégoire, Martin Juneau, François Simard, Philippe L’Allier, Anil Nigam, Josep Iglésies-Grau, Thomas Vincent, Deborah Talamonti, Emma Gabrielle Dupuy, Hânieh Mohammadi, Mathieu Gayda, Louis Bherer

**Affiliations:** 1Research Center and Centre ÉPIC, Montreal Heart Institute, Montréal, QC H1T 1N6, Canada; florent.besnier@umontreal.ca (F.B.); beatrice.berube@umontreal.ca (B.B.); malj@videotron.ca (J.M.); christine.gagnon@icm-mhi.org (C.G.); catherine-alexandra.gregoire@icm-mhi.org (C.-A.G.); martin.juneau@icm-mhi.org (M.J.); francois.simard.med@ssss.gouv.qc.ca (F.S.); philippe.lallier@icm-mhi.org (P.L.); anil.nigam@icm-mhi.org (A.N.); josep.iglesies-grau.med@msss.gouv.qc.ca (J.I.-G.); thomas.vincent@icm-mhi.org (T.V.); deborah.talamonti@gmail.com (D.T.); emma.dupuy@umontreal.ca (E.G.D.); hanieh.mohammadi@polymtl.ca (H.M.); louis.bherer@icm-mhi.org (L.B.); 2Department of Medicine, Université de Montréal, Montréal, QC H3C 3J7, Canada; 3Department of Psychology, Université du Québec à Montréal, Montréal, QC H3C 3P8, Canada; 4Research Center, Institut Universitaire de Gériatrie de Montréal, Montréal, QC H3W 1W5, Canada

**Keywords:** COVID-19, long COVID, rehabilitation, cognition, exercise, physical activity, respiratory rehabilitation

## Abstract

(1) Background: Cardiopulmonary and brain functions are frequently impaired after COVID-19 infection. Exercise rehabilitation could have a major impact on the healing process of patients affected by long COVID-19. (2) Methods: The COVID-Rehab study will investigate the effectiveness of an eight-week cardiopulmonary rehabilitation program on cardiorespiratory fitness ($\dot{V}$O_2_max) in long-COVID-19 individuals. Secondary objectives will include functional capacity, quality of life, perceived stress, sleep quality (questionnaires), respiratory capacity (spirometry test), coagulation, inflammatory and oxidative-stress profile (blood draw), cognition (neuropsychological tests), neurovascular coupling and pulsatility (fNIRS). The COVID-Rehab project was a randomised clinical trial with two intervention arms (1:1 ratio) that will be blindly evaluated. It will recruit a total of 40 individuals: (1) rehabilitation: centre-based exercise-training program (eight weeks, three times per week); (2) control: individuals will have to maintain their daily habits. (3) Conclusions: Currently, there are no specific rehabilitation guidelines for long-COVID-19 patients, but preliminary studies show encouraging results. Clinicaltrials.gov (NCT05035628).

## 1. Introduction

The clinical presentation of COVID-19 patients is varied and can include the following symptoms: fever, cough, breathlessness, anosmia, ageusia and fatigue. Most infected people develop a mild to moderate form of the disease and recover without hospitalization. Age and the presence of one or more comorbidities such as obesity or diabetes have an important part in the development and severity of the disease [1]. Systemic inflammation associated with the SARS-CoV-2 virus can induce cardiovascular and pulmonary sequelae such as thrombosis, myocarditis, diffuse alveolar damage or interstitial pulmonary fibrosis that appear to persist after the diagnosis of COVID-19 [2]. Some symptoms such as dyspnoea and fatigue appear to persist for several months after recovery in about 50% of cases [3,4]. Stress, anxiety, neurological and cognitive impairment have also been reported as long-term sequelae associated with the disease [2,3,4,5]. During post-hospitalization, Li et al. (2020) reported that 64% of patients still had sleep disturbances, 61% had poor exercise endurance, 58% had breathlessness, 62% had anxiety, and that 84% of them would have liked to receive advice on rehabilitation [6].

Physical deconditioning and reduced exercise capacity could be implicated in the general symptomatology [7,8,9]. Immobilization induced by hospitalizations and/or strict isolation at home exacerbate the clinical picture and symptoms, leading to a “deconditioning spiral” in these patients whose symptoms may persist long after the COVID-19 disease.

Cardiopulmonary rehabilitation is a cornerstone of the management of people affected by chronic pulmonary disease [10,11] and people with cardiovascular disease [12]. These rehabilitation programs are based on an individualised exercise-training program under supervision (aerobic exercises, muscle strengthening and breathing exercises). Beyond the benefit of exercise rehabilitation on morbi-mortality, an important amelioration of symptoms, cardiorespiratory fitness and quality of life due to pulmonary rehabilitation have also been reported [11]. The first randomised, controlled observational study on 72 patients diagnosed with COVID-19 demonstrated that a six-week pulmonary-rehabilitation program resulted in a significant improvement in respiratory functions (forced expiratory volume in one second (FEV1), forced vital capacity (FVC), Tiffeneau–Pinelli index), maximal distance on the 6 min walking test, quality of life, and anxiety [13]. Moreover, a case study in an 80-year-old female patient (severe COVID-19, 14 days of intensive care with intubation) reported similar results on walking ability [14]. In a recent study, an eight-week exercise-rehabilitation program combining aerobic and resistance exercises markedly increased cardiopulmonary and skeletal-muscle functions in patients with COVID-19 [15]. Several hospital departments, research teams and scientific societies have since made recommendations to implement pulmonary-rehabilitation programs for COVID-19 patients [16,17,18,19,20,21]. However, several questions remain about the beneficial impact of rehabilitation in patients with long COVID-19. In the rehabilitation program developed by Liu et al., breathing exercises were proposed but not aerobic exercise or peripheral-muscle strengthening [13]. Moreover, they did not assess cardiorespiratory fitness ($\dot{V}$O_2_ peak) or ventilatory responses during exercise. The cardiopulmonary-exercise test is extensively utilised in cardiac- and pulmonary-rehabilitation programs. It consists of an integrated exploration of respiratory, cardiovascular and metabolic functions. The $\dot{V}$O_2_ peak is the main prognostic marker of morbi-mortality [22]. The submaximal parameters allow for the quantification of the severity of the disease, the tracking of factors limiting exercise, and the evaluation of the mechanisms responsible of breathlessness [23], including in COVID-19 patients [24]. Finally, it allows for the individualization of the exercise-training program to each patient’s own training capacity and for the objectification of the effectiveness [12]. Furthermore, no data exist on the effect of a cardiopulmonary-rehabilitation program on brain health, including cognition and cerebral oxygenation, or on inflammatory and coagulation markers. At the cerebral level, recent studies suggest that people with long COVID-19 could have sequelae, even in the long term, and that cognitive deficits should be monitored [5]. A potential mechanism, the “inflammatory storm” induced by the SARS-CoV-2 virus, crosses the blood-brain barrier, implicating an alteration of brain functions by a cytokine shock that can even create silent strokes since it also causes an increase in coagulation markers [5,25].

There is no interventional study with a control group that has evaluated the effects of a cardiopulmonary-rehabilitation program on the evolution of the $\dot{V}$O_2_ peak in long-COVID-19 patients with persistent symptoms of breathlessness and chronic fatigue. In addition, an “integrative” analysis encompassing the complexity of long-COVID-19 symptoms, combining functional and respiratory tests, as well as assessments of the inflammatory and pro-oxidative profile, coagulation factors, quality of life, and cognitive and neurological functions, has not yet been performed.

The COVID-Rehab study will investigate the effectiveness of an eight-week cardiopulmonary-rehabilitation program on cardiorespiratory fitness ($\dot{V}$O_2_max) in long-COVID-19 individuals. Secondary objectives will include functional capacity, quality of life, perceived stress, sleep quality (questionnaires), respiratory capacity (spirometry test), coagulation, inflammatory and oxidative-stress profile (blood draw), cognition (neuropsychological tests), neurovascular coupling and pulsatility (fNIRS).

## 2. Materials and Methods

### 2.1. Study Design

The COVID-Rehab study will be a randomised clinical trial, with two intervention arms (1:1 ratio) that will be blindly evaluated: (1) rehabilitation: centre-based exercise-training program (8 weeks, 3 times per week); (2) control: individuals will have to maintain their daily habits. Participation in the study will last 10 weeks including 1 week of testing at baseline and 1 week post-intervention, in addition to the 8 training weeks.

### 2.2. Participants

A total of 40 participants with long COVID-19 will be included.

Inclusion criteria.

Participants will be adults aged between 40 and 80 years old. Individuals will be included if they have had a positive polymerase-chain-reaction (PCR)-test diagnosis of the SARS-CoV-2 virus and are still suffering from dyspnoea and/or fatigue >3 months after the diagnosis of COVID-19, or if they still have an increase of 1 point in dyspnoea on the Modified Medical Research Council dyspnoea scale compared to the pre-infection period. All participants must have no contra-indication to exercise testing and training [12] and must be able to read, understand and sign the information and consent form.

Exclusion criteria.

Individuals with one of the exclusion criteria will not be eligible for our research project:Pulmonary embolism diagnosed by scintigraphy.Absolute and relative contraindication to cardiopulmonary stress test or exercise training [12].Severe exercise intolerance, significant cardiac arrhythmia or ischemia during low-intensity exercise, severe pulmonary hypertension.Severe pulmonary disease (e.g., chronic obstructive pulmonary disease, severe COVID-19-related symptoms, severe asthma).Recent cardiovascular events (cardiac decompensation, angioplasty or cardiac surgery less than 4 weeks; valvular heart disease requiring surgical correction, myopericarditis, unstable ventricular rhythm disturbances despite treatment).Kidney failure requiring dialysis.Heart failure (NYHA III or IV).

### 2.3. Interventional Methods

Recruitment will be ensured by a pulmonologist who will carry out the pre-inclusion process and will verify participants’ eligibility. The research coordinator will then call the pre-selected participants to explain the research project. If they show interest in the study, potential participants will receive the consent form by email. They will have a few days before the coordinator schedules the first visit. During the first visit, the research team will give an explanation and respond to the participant’s questions, and participants will sign the information and consent form. Additionally, during the first visit, participants will have an appointment with a cardiologist from the Montreal Heart Institute’s Centre ÉPIC, who will confirm eligibility criteria and then prescribe the rehabilitation program.

#### 2.3.1. Measurements and Outcomes

Three assessment visits will be made prior to the intervention, as well as at the end of the intervention. All the testing procedures are reported in Table 1. At baseline, demographic and clinical characteristics (previous diseases, comorbidities, medication) will be assessed by the research staff.

#### 2.3.2. Primary Outcome: Cardiorespiratory Fitness

The primary outcome of the study will be cardiorespiratory fitness ($\dot{V}$O_2_ peak: mL/kg/min) measured by a maximal cardiopulmonary-exercise test (CPET) on an ergocycle (Ergoline 800S, Bitz, Germany). The protocol will be individualised and will include a 3 min warm up at 20 W, followed by an increase in workload from 10 to 20 W/min (depending on the participant’s fitness level) until exhaustion, while maintaining cadence >60 rpm. Electrocardiogram (Marquette, case 12, St. Louis, MI, USA) and oxygen saturation will be continuously monitored, whereas the rating of perceived exertion (Borg Scale, 6–20) and manual blood pressure (sphygmomanometer: Welch Allyn Inc., Skaneateles Falls, NY, USA) will be measured every 2 min during the test. Participants will be encouraged during the test. Minute ventilation ($\dot{V}$E: L/min), oxygen uptake ($\dot{V}$O_2_: mL/min) and carbon dioxide production ($\dot{V}$CO_2_: mL/min) will be continuously measured at rest as well as during exercise and recovery using a metabolic system (Cosmed Quark, Rome, Italy). Gas exchange will be collected on a breath-by-breath basis and expressed with a 15 s time averaging for analysis. The highest $\dot{V}$O_2_ value during the exercise period will be considered as the $\dot{V}$O_2_ peak.

#### 2.3.3. Secondary Outcomes: Functional and Respiratory Capacity, Quality of Life

Functional capacity will be assessed with the 6 min walking test (6MWT), the Timed Up-and-Go test (TUG) and the Sit-to-Stand test (5-STS). The 6MWT is a validated sub-maximal exercise test widely used in pulmonary-rehabilitation programs to assess aerobic capacity in adults and elderly adults [10]. The aim of the 6MWT is to cover the highest walking distance possible in 6 min on a 30 m walkway. The maximum distance walked is recorded. Heart rate, oxygen saturation and rating of perceived exertion (Borg Scale, 6–20) will be measured at rest, during the test and just after the end of the test, during the recovery period. During the TUG, the time taken by the participant to get up from the chair, walk 3 m, turn around, and sit back in the chair will be recorded.

This test has been used for the assessment of functional mobility, walking ability, dynamic balance, and risk of falling in subjects with a variety of conditions including COPD [26]. It will be performed at spontaneous and quickest velocities. The 5-STS test (i.e., five stand-ups and sit-downs in a row performed as quickly as possible without using arms) is used in rehabilitation programs and in healthy older community-living populations [27]. The test is reliable, valid, and partially dependent on leg-muscle force and balance [28].

Respiratory capacity will be assessed with spirometry in a sitting position (Cosmed Quark, Rome, Italy) according to the American Thoracic Society/European Respiratory Society recommendation [11]. Spirometric measurements will include: the forced expiratory volume in 1 s (FEV1); the forced vital capacity (FVC), which is the maximal amount of air that can be forcibly exhaled from the lungs after full inspiration; the vital capacity (VC), which is the maximum amount of air that can be exhaled when blowing out as fast as possible; the FEV1/FVC ratio; the peak expiratory flow (PEF) is the maximal flow that can be exhaled when blowing out at a steady rate; the forced expiratory flow (with the rates at 25%, 50% and 75% FVC); the inspiratory vital capacity (IVC) is the maximum amount of air that can be inhaled after a full expiration. Regular (normal) breaths will first be taken with the mouthpiece in place, then a deep, full breath, followed by a quick, full exhale to complete the exam. Three acceptable manoeuvres will be achieved. The best of the three trials will be considered.

Several questionnaires will be used to measure quality of life (36-Item Short-Form Health Survey), stress (State-Trait Anxiety-Inventory questionnaire, Perceived-Stress Scale questionnaire), sleep quality (Pittsburg Sleep-Quality Index questionnaire), perceived health, well-being and daily life of patients with obstructive-airway symptomatology (adapted St. George’s Respiratory Questionnaire), range of respiratory disability (Medical Research Council Breathlessness Scale) and symptoms associated with COVID-19 (adapted from “WHO’s Global COVID-19 Clinical Platform: Post COVID-19 CRF” [29] and “Long Coronavirus Disease (COVID) Symptom and Impact Tools: A Set of Patient-Reported Instruments Constructed From Patients’ Lived Experience” from Tran et al. [30]).

#### 2.3.4. Tertiary Outcomes: Coagulation, Inflammatory and Antioxidant Profile and Brain Health

Criteria for coagulation include: complete blood count, PTT (thromboplastin time, in sec), Fibrinogen (factor I in g/L), PT (prothrombin level, in sec and in%), and INR (international normalised ratio). The criteria for evaluating the inflammatory profile include: C-reactive protein, TNF alpha, IL-6, IL-1, IL-10, lactate dehydrogenase (LDH). Those of the pro/antioxidant balance include: uric acid, albumin, myeloperoxidase (MPO).

Brain health: the neuropsychological evaluation will be carried out by videoconference. Participants will be tested for general cognitive function using a remote version of the Montreal Cognitive Assessment (MoCA) [31]. The neuropsychological tests will be performed in a fixed order and will assess different components of cognition: the Hopkins Verbal Learning Test (verbal memory), the Digit Span (short-term memory and working memory), an oral version of the Trail-Making Test (executive functions) [32], as well as a phonological and semantic verbal-fluency test (language and executive functions). The neuropsychological tests are validated for remote administration and have been used previously in an important cohort [33]. In addition, these tools are validated and normalised for subjects of 40 years and over [34]. Before the evaluation, we will make sure that the participants have the necessary technological tools and adequate internet connections in order to carry out the evaluation in an appropriate setting.

Neurovascular coupling and cortical-pulsatility index will be measured using functional near-infrared spectroscopy (fNIRS). fNIRS is a non-invasive optical-imaging technique that measures changes in haemoglobin (Hb) concentrations within the brain by means of the characteristic absorption spectra of Hb in the near-infrared range. The fNIRS system will comprise 23 channels placed over the prefrontal cortex. The first fNIRS session will consist of a 5 min baseline recording from which a pulsatility index will be extracted. The second fNIRS session will involve a dual task: the working-memory task while walking at a normal pace. The working-memory task will consist of a 2-back task, where the series of numbers is communicated by a headset. Every time a participant hears a number, they will be asked to repeat the number they heard two positions before.

### 2.4. Cardiopulmonary-Rehabilitation Program

The cardiopulmonary-rehabilitation program will consist of aerobic exercise, muscle strengthening, and respiratory exercise 3 times per week over 8 weeks. The program will be individualised according to the F.I.T.T. principles (frequency, intensity, time, and type) of the respiratory- and cardiovascular-rehabilitation guidelines [10,12]. All sessions will be supervised by a kinesiologist from the Montreal Heart Institute’s Centre ÉPIC. Aerobic exercise will be performed on a cyclo-ergometer and will last 30 min maximum (with a 5 min warm-up and cool-down). The intensity will be individualised at the first ventilatory threshold according to the cardiorespiratory-exercise test and may be gradually increased depending on the patient’s tolerance. Heart rate, oxygen saturation and perceived level of exertion (Borg scale of 6–20, with a target of 11 to 13) will be measured for each exercise session. Then, after a 10 to 15 min break, strengthening exercises will be performed with weight machines, free weights and/or elastic bands, and will target large muscle groups (3 sets of 10 repetitions, starting at an intensity of 40% of the maximum strength or at 4 to 5 on the OMNI scale 0–10 for resistance training [35]). The session will end with respiratory exercises including two types of exercises: (1) pursed-lip abdominal breathing exercise: in supine position with legs bent and a weight of 1 to 3 kg placed on the lower abdomen to create resistance for the contraction of the diaphragm. The patient will be instructed to inhale through the nose and inflate the stomach, then to slowly exhale through the mouth with pursed lips (3 sets of 10 cycles). (2) inspiratory muscle training with the POWERbreathe^®^ device. This tool is a system with a valve that creates a resistance and that can be adjusted to train the respiratory muscles. Patients will perform 3 sets of 10 breathing cycles. They will be asked to breathe into the device at a steady pace, increasing their breathing.

### 2.5. Research Plan

Following the pre-selection, 3 visits will be made with the subjects. The first visit will start with the project being re-explained to the participant, and by the participant signing the information and consent form. The first session will also include a medical visit with a pulmonologist or a cardiologist who will ensure that participants meet the eligibility criteria. The anamnesis will retrace the medical history and the comorbidities, as well as the results of previous medical examinations and the treatments already prescribed. Then, a nurse will perform a blood draw (a total of 28 mL). A cardiologist will supervise the maximal cardiopulmonary exercise testing. Prior to this test, a spirometry will assess respiratory function. All the questionnaires will be explained to participants who will later answer them from home. The neuropsychological evaluation will be carried out by videoconference during the second session; a neuropsychologist or a trained research assistant will conduct the neuropsychological assessment. The third assessment session will include the brain-imaging fNIRS session and the functional tests. The 3 sessions will also be performed at the end of the intervention in the same order. Total duration of the study is 10 weeks, including 8 weeks of intervention (Table 1).

## 3. Data analysis

### 3.1. Sample-Size Calculation

Sample size for primary outcome was performed by a biostatistician from the Montreal Health Innovations Coordinating Centre (MHICC). This calculation was performed using previously published values in the literature of $\dot{V}$O_2_ peak in COVID-19 individuals [36,37,38,39] and the effect of a six-week rehabilitation program on the 6 min walking test performance [13]. A sample size of 17 subjects in each group will have 80% power to detect a difference in means of 3 mL/min/kg (difference between control-group mean of 19 mL/min/kg and cardiopulmonary-rehabilitation-group mean of 22 mL/min/kg, assuming that the common standard deviation is 3 using a two-group *t*-test (0.050 two-sided significance level). As we expect a 15% attrition rate based on our previous exercise studies, we will thus recruit 20 participants per intervention arm, for a total of 40 individuals with long COVID-19. A biostatistician from the MHICC will generate the randomization sequence.

### 3.2. Statistical Analysis

The variables in the study will be presented using descriptive statistics. The mean, standard deviation, median, minimum, Q1, Q3, and maximum will be presented for continuous variables. The number and percentage will be presented for nominal/ordinal variables. The assumptions of the statistical tests will be examined, and data transformation or non-parametric analyses may be used as appropriate. STATA SE software version 15 will be used to conduct the analyses (STATA SE 15, StataCorp LP, College Station, TX, USA). Mean changes in cardiorespiratory fitness ($\dot{V}$O_2_ peak) from baseline will be analysed using a covariance-analysis model (ANCOVA) including baseline value and group intervention. Secondary and tertiary outcomes will be analysed as the primary outcome. No imputation will be used for missing values, and analyses will be performed on the intention-to-treat principle (according to the group assigned by randomization). All statistical tests will use a two-tailed significance level of 0.05.

### 3.3. Blinding

Assessors performing the evaluations and investigators will be blinded to group allocation. The statistician will be blinded until completion of the statistical analyses. Only the kinesiologists executing the cardiopulmonary-rehabilitation program will be aware of the assigned intervention. Kinesiologists will not take part in any assessment.

## 4. Discussion

COVID-19 infection may induce abnormalities in most of the physiological systems including the respiratory, cardiovascular and neurological systems. Symptoms such as breathlessness and fatigue seem to last for several months after recovery in about 50% of cases. Stress, anxiety, neurological and cognitive impairment have also been reported in the long-term sequelae. CPET is the gold standard by which to measure the effectiveness of the O_2_-transport chain (pulmonary, cardiac, muscle and vascular system). Previous studies on post-COVID-19 patients show a moderate impairment of their physical capacity, probably caused by muscle deconditioning [37,39]. In addition, the immobilization induced by hospitalizations and/or strict isolation at home promotes a spiral of deconditioning with a massive increase in sedentary/inactive time [40].

To counteract the significant disability of individuals with long COVID-19 and the deterioration of their quality of life, cardiopulmonary-rehabilitation programs have begun to be recommended in the care of patients after COVID-19 [13,16,17,41]. In 2020, the Pan American Health Organization associated with the World Health Organization published guidelines in this direction, both for patients who seriously suffer from the disease in the acute phase and for those who continue to suffer from the long-term consequences of COVID-19 [21]. Individualised rehabilitation-training programs including aerobic exercises, muscle strengthening and specific breathing exercises could allow people to recover better.

Nevertheless, some cautions have to be taken. Some individuals with long COVID may additionally experience post-exertional malaise after physical activity [42]. The worsening of symptoms after exercise may include fatigue or exhaustion, cognitive dysfunction, pain, and sleeping disturbance. In patients with long COVID-19 (*n* = 3762; from 56 countries), the main reported symptoms after six months were fatigue, post-exertional malaise, and cognitive dysfunction [42]. Relapses, mainly triggered by physical exercise, mental activity and stress, were observed in 85.9% of individuals (95% CI, 84.8% to 87.0%) [42]. The second version of the recommendations for the clinical management of people with COVID-19 from the WHO provides guidance on rehabilitation [18]: returning to activities of daily living is therefore a priority but must be done at an appropriate, safe and individualised pace within the limits of the symptoms. Exercise intensity should not be pushed because of the risk of post-exercise fatigue. A gradual increase in exercise should be based on symptoms [18]. Altogether, currently, specific rehabilitation guidelines for long-COVID-19 patients are not available. Studies are needed to assess the effectiveness of programs and their safety [43,44]. The expertise of cardiopulmonary-rehabilitation services should also be considered [45].

### Study Limitations

It seems that an significant portion of patients have a reduced $\dot{V}$O_2_ peak after COVID-19 compared to controls (around 30%). However, the “cardiorespiratory-exercise” profiles seem quite varied and also include patients who do not report a reduction in their performance (in comparison with the theoretical values according to age) [37]. On the other hand, the effect of cardiopulmonary-rehabilitation programs on long-COVID-19 patients is under investigation, but data are not yet available [43]. In total, the calculation of the required number of subjects is intricate. Secondly, considering that some patients report an exacerbation of post-exercise symptoms and that our program is relatively short (two months), maybe a longer program could be needed for these patients. The optimal conditions of a safe and effective rehabilitation program remain to be determined (types of activity, frequency, duration, intensity). Finally, the study is monocentric and will possibly limit the generalization of the results.

## 5. Conclusions

Cardiopulmonary and brain functions are frequently impaired after COVID-19 infection. Physical deconditioning and reduced exercise capacity could be implicated in the general symptomatology. Cardiopulmonary rehabilitation is the cornerstone to the management of people affected by chronic pulmonary and cardiovascular diseases and may be relevant to individuals living with long COVID-19. Studies are needed to assess the effectiveness of rehabilitation programs and their safety in this population.

## Figures and Tables

**Table 1 ijerph-19-04133-t001:** Enrolment, assessments and interventions according to SPIRIT guidelines.

Time Point	t−_1_	T_0_ Baseline Evaluations			T_1_ 10 Weeks
		Visit 1	Visit 2	Visit 3	Visit 4, 5, 6
Enrolment:					
Eligibility screening	X				
Informed consent	X				
Assessments:					
Medical visit		X			X
Blood draw ^1^		X			X
Body-composition analysis		X			X
Spirometry		X			X
$\dot{V}$O_2_max		X			X
Neuropsychological assessment ^2^			X		X
NIRS pulsatility, neurovascular coupling				X	X
5 Sit-to-Stand test				X	X
Timed Up-and-Go test				X	X
6 min walking test				X	X
Self-reported questionnaires ^3^				X	X
Interventions:					
control group				Have to maintain their daily habit	
cardiopulmonary-exercise-training group				8 weeks; 3 times/week	

^1^ complete blood count, PTT (thromboplastin time), Fibrinogen (factor I in g/L), PT (prothrombin level, in sec and in %), INR (international normalised ratio), C-reactive protein, TNF alpha, IL-6, IL-1, IL-10, lactate dehydrogenase (LDH), Uric acid, Albumin, Myeloperoxidase (MPO). ^2^ remote version of the Montreal Cognitive Assessment (MoCA), Hopkins Verbal Learning Test (verbal memory), Digit Span (short-term memory and oral-memory version of the Trail-Making Test (executive functions), phonological and semantic verbal-fluency test (language and executive functions). ^3^ SF-36 (quality of life), stress (State-Trait Anxiety-Inventory questionnaire, Perceived-Stress Scale questionnaire), sleep quality (Pittsburgh Sleep-Quality Index questionnaire) and long-COVID-19 symptoms questionnaire.

## Data Availability

To protect the integrity of the major objectives of COVID-Rehab, data that break the blind will not be presented prior to the release of the main results. This release will be announced by the sponsor-investigator of the study. Authorship of the publication reporting the main outcome as well as the ancillary studies will be defined in compliance with the recommendations of the International Committee of Medical Journal Editors. The results of the study will also be released to the participants, the Centre ÉPIC staff, and more broadly to the general medical community. Finally, no later than 1 year after COVID-Rehab end (i.e., completion of the final assessment by the last participant), the data of COVID-Rehab study will be available for sharing purposes from the principal investigator under reasonable request.

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
