# Peer review of "Cardiopulmonary Rehabilitation in Long-COVID-19 Patients with Persistent Breathlessness and Fatigue: The COVID-Rehab Study"

_ijerph, 2022, doi:10.3390/ijerph19074133_

Round 1

Reviewer 1 Report

If there is a previous study by the first author regarding this study, it is necessary to explain the methods which was used. The rehabilitation for the physical function after the COVID-19 is an important issue in the future.

Since the test itself used in this study is used a common test clinically, it will be originality by showing the clinical data of patients suffering from COVID-19.

In addition, since the content of rehabilitation is also data of patients suffering from COVID-19, it seems that original ingenuity is required, but this paper cannot be analyzed and discussed without the result data.

Therefore, it is difficult to evaluate as peer review for me because there is no the result’s data used for this disease. From the above, preliminary experimental data based on the method related to this case is required. If preliminary study has been performed, all data values used in this study should be explained.

Author Response

R1 : If there is a previous study by the first author regarding this study, it is necessary to explain the methods which was used. The rehabilitation for the physical function after the COVID-19 is an important issue in the future.

We would like to thank reviewer 1 for this comment. Even if our group has experience in cardiovascular and cardiopulmonary rehabilitation, we don’t have previous experience and we have never published a study on long COVID-19 patients specifically. Thus, methods used for rehabilitation are the same than in cardiac patients or other clinical pathologies, as follows: 1/ cardiopulmonary assessments, 2/ individualisation of exercise training sessions based on the cardiopulmonary exercise stress test, and the patient’s feelings and level of energy/long COVID symptoms. 3/ a program combining aerobic exercises, muscle strengthening and respiratory exercises. Barbara et al. (Eur J Prev Cardiol, 2022) has recently published encouraging results in long COVID-19 patients with a similar program.

Since the test itself used in this study is used a common test clinically, it will be originality by showing the clinical data of patients suffering from COVID-19.

Unfortunately, we do not have data from long COVID-19 patients. In the literature, it seems that individuals with long COVID-19 have an average VO2peak of 80% +/-10 of predicted values. But data in the literature is still scare regarding VO2 values in long COVID patients

In addition, since the content of rehabilitation is also data of patients suffering from COVID-19, it seems that original ingenuity is required, but this paper cannot be analyzed and discussed without the result data. Therefore, it is difficult to evaluate as peer review for me because there is no the result’s data used for this disease. From the above, preliminary experimental data based on the method related to this case is required. If preliminary study has been performed, all data values used in this study should be explained.

This manuscript is a protocol paper for a pilot study. Thus, we unfortunately do not have data to present at this point. Further, we have never published data in long COVID-19 patients. Nevertheless, you can find some other pilot studies, as the one by Barbara et al. (Eur J Prev Cardiol, 2022), who reported an improvement of +2.7 mL.kg.min after an 8-week rehabilitation program. https://doi.org/10.1093/eurjpc/zwac019  all abbreviations.

Reviewer 2 Report

  Your paper describes the protocol for a randomized controlled trial of the effects of cardiopulmonary rehabilitation on aerobic capacity and other clinical variables in patients with post COVID-19 syndrome.  Given the prevalence of post COVID syndrome and the lack of definitive evidence of the benefits of exercise training in this condition, your study is  very important.

I have a few suggestions:

  1.  How will  potential subjects be identified?  Will you rely on patients who seek care at a post COVID care clinic?  Will you market the protocol among primary care providers, pulmonologists and cardiologists, etc.?
  2.   You describe the power calculation for determination of the number of subject required  based on a change in VO2 peak of 3 milliliters/kilogram per minute.  However, you need to explain why you chose an 8 week rehabilitation program rather than a standard 12 week rehabilitation program.
  3. Patient's with persistent breathlessness and fatigue will be recreated as subjects.  Does this mean that patients with orthostatic intolerance/tachycardia or other prominent symptoms will be excluded?  Will you exclude patient's who have an above average aerobic capacity at baseline and appear not to be impaired from a cardiorespiratory fitness perspective?  How is severe exercise intolerance (an exclusion criterion) defined?
  4. Lines 72 and 162: change "testing" to "test".
  5. Be consistent with terminology: use "fatigue" rather than "asthenia".
  6. More detail regarding the prescription of aerobic exercise training intensity is needed since this is a critically important variable in increasing cardiorespiratory fitness with training. Will you use a target heart rate or perceived exertion range? Will high intensity interval training be incorporated into the training protocol or will training include moderate intensity continuous training only?
  7. You will measure many important clinical variables pre/post rehabilitation. Consider adding change in skeletal muscle strength as an additional fitness variable.

Author Response

R2: Your paper describes the protocol for a randomized controlled trial of the effects of cardiopulmonary rehabilitation on aerobic capacity and other clinical variables in patients with post COVID-19 syndrome.  Given the prevalence of post COVID syndrome and the lack of definitive evidence of the benefits of exercise training in this condition, your study is very important.

Thank you very much for the opportunity to review the study „Cardiopulmonary rehabilitation in long COVID-19 patients with persistent breathlessness and fatigue: the COVID-Rehab study“ by Florent Besnier et al.

We would like to thank you for your revision and comments. It helps to improve the quality of the manuscript.

I have a few suggestions:

  1. How will  potential subjects be identified?  Will you rely on patients who seek care at a post COVID care clinic?  Will you market the protocol among primary care providers, pulmonologists and cardiologists, etc.?

Patients will be identified by our team’s pulmonologist (listed in the authors of the manuscript), who will overview the history of COVID-19 disease, as well as the eligibility criteria.

  1. You describe the power calculation for determination of the number of subject required based on a change in VO2 peak of 3 millilitres/kilogram per minute.  However, you need to explain why you chose an 8-week rehabilitation program rather than a standard 12-week rehabilitation program.

We thank the reviewer for this relevant comment, which was also a matter of debate in our group. The optimal ‘dose’ of rehabilitation sessions is still unclear in populations with chronic disease. Our approach is based on individualization, which includes 24 sessions, 3x per week for 8 weeks. We believe that 8 weeks is the minimum necessary to induce a clinically significant improvement in cardiopulmonary capacity in long COVID-19 individuals. Of course, potentially 3 months or more could provide better results. For example, in a recent original article, Barabara et al. (Eur J Prev Cardiol, 2022) reported an improvement of +2.7 mL.kg.min after an 8-week rehabilitation program in long-COVID19 individuals. https://doi.org/10.1093/eurjpc/zwac019. In the context of a pilot study and for feasibility purposes, we chose an 8-week training period.

  1. Patient's with persistent breathlessness and fatigue will be recreated as subjects.  Does this mean that patients with orthostatic intolerance/tachycardia or other prominent symptoms will be excluded?  Will you exclude patient's who have an above average aerobic capacity at baseline and appear not to be impaired from a cardiorespiratory fitness perspective?  How is severe exercise intolerance (an exclusion criterion) defined?

To clarify our point, patients with orthostatic intolerance/tachycardia will also be recruited if they also experience persistent breathlessness and/or fatigue. Individuals with a VO2peak value in a normal range (90-110% of the theoretical value) will also be recruited if they experience persistent breathlessness and/or fatigue. One limit of our study is the unavailability of participants’  physical fitness prior to SARS-COV-2 infection. It is possible that some participants might have had a VO2max value higher than 100% prior to infection.

The definition of exercise intolerance is reported by the ACSM’S guidelines for exercise testing and prescription. Signs/symptoms of exercise intolerance include the following: signs or symptoms of ischemia, excessive dyspnea or fatigue at a low level of exercise intensity, dizziness or lightheadedness, inotropic or chronotropic incompetence, low oxygen saturation (< 85%) with light exercise. Each of these examples could lead to further medical testing before starting any rehabilitation program. A cardiologist or a pulmonologist will provide their approval for each participant before their beginning in the study and will supervise each maximal cardiopulmonary exercise test.

  1. Lines 72 and 162: change "testing" to "test".
  2. Be consistent with terminology: use "fatigue" rather than "asthenia".

Thank you for pointing out this vocabulary mistake. We have made changes to the text.

  1. More detail regarding the prescription of aerobic exercise training intensity is needed since this is a critically important variable in increasing cardiorespiratory fitness with training. Will you use a target heart rate or perceived exertion range? Will high intensity interval training be incorporated into the training protocol or will training include moderate intensity continuous training only?

Thank you for your suggestion. We would like to underline that one of the reviewers suggest to present these technical details in a “supplementary materials” as this part is too long. To precise here the program, as this is a pilot study with individuals living with long-COVID-19, we adopted a conservative approach with only moderate and continuous training targeting the first ventilatory threshold. With a proportion of patients experiencing an exacerbation of their symptoms, we are not comfortable to systematically propose a protocol with HIIT (although interval exercise should be proposed, high intensity should be avoided). Firstly, we will use perceived exertion range (11 to 13/20 in the Borg Scale) while monitoring heart rate and power (watts). The first session will begin at 40% of peak power measured at the maximal exercise test. Then after 1 week, the training power could be increased at ventilatory threshold 1, while respecting the perceived exertion range, heart rate AND the symptoms experienced in the 24h after each session.

  1. You will measure many important clinical variables pre/post rehabilitation. Consider adding change in skeletal muscle strength as an additional fitness variable.

We do not measure the strength neither at the diaphragm level (with sniff nasal inspiratory pressure test - SNIP test), nor at the peripheral level with a dynamometer. Only the Sit-to-Stand test will be used. It is a validated method to estimate (not a measurement) the strength of the lower limbs. The measurement of the 1-RM was debated, but we chose to train patients with the OMNI scale (0-10) for resistance training with a target between 4 and 5.

Reviewer 3 Report

Review „Cardiopulmonary rehabilitation in long COVID-19 patients with persistent breathlessness and fatigue: the COVID-Rehab study“ by Florent Besnier et al

Thank you very much for the opportunity to review the study „Cardiopulmonary rehabilitation in long COVID-19 patients with persistent breathlessness and fatigue: the COVID-Rehab study“ by Florent Besnier et al.

In their work, Besnier and colleagues will investigate the effectiveness of an 8-week cardiopulmonary rehabilitation program on cardiorespiratory fitness (VO2max) in long COVID-19 individuals, as there are currently no specific rehabilitation guidelines for long COVID-19 patients but preliminary studies showing encouraging results.

The study is designed as a prospective, single blind (evaluators), randomized trial with two parallel intervention arms (1:1) recruiting a total of 40 individuals either randomized to rehabilitation or to control (having to maintain their daily habits). Secondary objectives will include functional capacity, quality of life, perceived stress, sleep quality (questionnaires), respiratory capacity (spirometry test), coagulation, inflammatory and oxidative stress profile (blood draw), cognition (neuropsychological tests), neurovascular coupling and pulsatility (fNIRS).

The paper is well written and below you will find some minor points of criticism.

M&M section

Participants

How did you generate the sample size calculation? Consider lifting the explanation given below up a little further.

The authors should be some more precise in the exclusion criteria, e. g. “Pulmonary embolism diagnosed by scintigraphy” – why is that?

Is data protection fully considered?

Line 194: “Rome, Italie” – either “Italy” or “Italia”

Line 273: “The first session will also include. a“ – point too much.

After all, the authors as Canadians are native speakers, thus English language is fine.

I wish the authors much success with the study.

Author Response

R3: In their work, Besnier and colleagues will investigate the effectiveness of an 8-week cardiopulmonary rehabilitation program on cardiorespiratory fitness (VO2max) in long COVID-19 individuals, as there are currently no specific rehabilitation guidelines for long COVID-19 patients but preliminary studies showing encouraging results. The study is designed as a prospective, single blind (evaluators), randomized trial with two parallel intervention arms (1:1) recruiting a total of 40 individuals either randomized to rehabilitation or to control (having to maintain their daily habits). Secondary objectives will include functional capacity, quality of life, perceived stress, sleep quality (questionnaires), respiratory capacity (spirometry test), coagulation, inflammatory and oxidative stress profile (blood draw), cognition (neuropsychological tests), neurovascular coupling and pulsatility (fNIRS).

 The paper is well written and below you will find some minor points of criticism. After all, the authors as Canadians are native speakers, thus English language is fine. I wish the authors much success with the study.

M&M section

Participants

How did you generate the sample size calculation? Consider lifting the explanation given below up a little further.

The calculation was based on existing values available in the literature for cardiorespiratory fitness (VO2peak) and the 6-min walking test performance in individuals living with long COVID-19. For example, in a recent original article, Barbara et al. (Eur J Prev Cardiol, 2022) reported an improvement of +2.7 mL.kg.min after an 8-week rehabilitation program in long COVID-19 individuals. https://doi.org/10.1093/eurjpc/zwac019.

Based on these results and our experience in rehabilitation, the calculation revealed that a sample size of 17 subjects in each group will have 80% power to detect a difference in means of +3 mL.kg.min (difference between control group mean of 19 mL.kg.min and cardiopulmonary rehabilitation group mean of 22 mL.kg.min, assuming that the common standard deviation is 3 using a two-group t-test (0.050 two-sided significance level). As we expect a 15% attrition rate based on our previous exercise studies, we will thus recruit 20 participants per intervention arm, for a total of 40 individuals with long COVID-19.

The authors should be some more precise in the exclusion criteria, e. g. “Pulmonary embolism diagnosed by scintigraphy” – why is that?

The safety of exercise after a pulmonary embolism is still matter of debate. Recent data suggest that light exercise intensity is safe, but the literature is very recent on this subject (PE and exercise) and we believe that it should be treated separately. We therefore chose not to include patients who had a pulmonary embolism.

Is data protection fully considered?

All the data are anonym and will be encoded with a study I.D. number. This study is conducted in compliance with International Conference on Harmonization Good Clinical Practice (ICH-GCP) and all applicable regulatory requirements.

Line 194: “Rome, Italie” – either “Italy” or “Italia”

Line 273: “The first session will also include. a“ – point too much.

We thank the reviewer for pointing out these mistakes. We have made changes to the text accordingly.

Reviewer 4 Report

The manuscript represents the protocol of an already started prospective, randomized, controlled and single blinded study to evaluate the effect of cardiopulmonary rehabilitation (CPR) on cardiorespiratory fitness in patients suffering from long COVID-19 syndeome. CPR ist projected to take 8 weeks in total, thereby performing three CPR sessions per week.

Comments:

  1. To adapt CPR to the special needs of post COVID patients is certainly an important issue. From this standpoint of view it would be of great public interest to find a scientifically and ethical correct way to extend this study by prolonging the follow-up period to et least 1 year or even longer. Moreover, the study could be strengthened by including additional rehailitation centres.
  2. CPR programs deliver a combination of interventions including medical issues, individually adapted physical exercise, psycological and social support, a well as information, motivation and education (see DACH guidelines JCM 2021 and others). This multimodal therapeuthic concept might especially be important in patients with "long COVID-19", and therefore should be presented in a more transparent way.  For example it would be of interest to know about the diagnostic and therapeutic consequences, if some of the diagnostic parameters are suspicious. Especially, it is of interest to know more about potential psychological and psychosocial interventions.
  3. The outlay of the present manuscript ist not satisfactory, and reading the long text passages is tiring and not attractive even for the interested clinician. Therefore it is recommended to especially shorten the "Introduction" and the "Discussion" sections
  4. The methods strictly should follow the "PICO`s" system clearly describing the "population" under investigation, the "interventions", the "controls" and the anticipated "outcomes". Please use Tables to help the reader by catching quick overviews on text and contents!
  5. Please also present a flow chart which outlines the various steps of the study protocol. This flow-chart should replace Table 1, the technical outlay of which is not acceptable.
  6. The therapeutic interventions during CPR are actually outlined in long text passages. Again, all these interventions should preferably be presented in a table, and technical details should be presented in "supplemental materials".
  7. All abbreviations should be listed at the beginning or at the end of the text.

Author Response

R4: The manuscript represents the protocol of an already started prospective, randomized, controlled and single blinded study to evaluate the effect of cardiopulmonary rehabilitation (CPR) on cardiorespiratory fitness in patients suffering from long COVID-19 syndeome. CPR ist projected to take 8 weeks in total, thereby performing three CPR sessions per week.

Comments:

  1. To adapt CPR to the special needs of post COVID patients is certainly an important issue. From this standpoint of view it would be of great public interest to find a scientifically and ethical correct way to extend this study by prolonging the follow-up period to at least 1 year or even longer. Moreover, the study could be strengthened by including additional rehabilitation centres.

We would like to thank the reviewer for these suggestions. Multicentre studies and long-term follow- ups would be the best way to validate the effects of the CPR program. However, this additional follow-up and the inclusion of other centres has a cost. We understand the limitations of our pilot study and we recognize that the generalizability of the data will be limited by this. We added some sentence in the study limitation section.

  1. CPR programs deliver a combination of interventions including medical issues, individually adapted physical exercise, psycological and social support, a well as information, motivation and education (see DACH guidelines JCM 2021 and others). This multimodal therapeuthic concept might especially be important in patients with "long COVID-19", and therefore should be presented in a more transparent way.  For example it would be of interest to know about the diagnostic and therapeutic consequences, if some of the diagnostic parameters are suspicious. Especially, it is of interest to know more about potential psychological and psychosocial interventions.

We agree with the importance of multidisciplinary care and interventions. While the study is centred on exercise therapy, our group is also specialized in neuropsychological assessments and follow-ups. During the baseline test or during the course of the project, if any sign of cognitive impairment is detected the patient will be taken care of by the neuropsychologist and may be referred to a neurologist if deemed necessary.

Our team also includes kinesiologists that are specialized with clinical populations. More than monitoring physical activity, they also provide motivational talks with patients to reassure and encourage them. However, we do not have dedicated psychological and psychosocial interventions in our study.

  1. The outlay of the present manuscript ist not satisfactory, and reading the long text passages is tiring and not attractive even for the interested clinician. Therefore it is recommended to especially shorten the "Introduction" and the "Discussion" sections.

We are sorry that the reviewer did not have a good reading experience of our manuscript. We would like to accommodate and include comments from all the reviewers and the editor. The discussion is less than one page long and the introduction is also one page, which seems reasonable to us. One of the reviewers and the academic editor pointed out that the paper is well written. We remain available to review the drafting of the parts if necessary after a second round.

  1. The methods strictly should follow the "PICO`s" system clearly describing the "population" under investigation, the "interventions", the "controls" and the anticipated "outcomes". Please use Tables to help the reader by catching quick overviews on text and contents!
  2. Please also present a flow chart which outlines the various steps of the study protocol. This flow-chart should replace Table 1, the technical outlay of which is not acceptable.

We would like to thank you for your suggestion. There are different ways to present a research protocol. We chose the most recommended and used method (SPIRIT).

PICO’s system is a mnemonic tool like any other to synthesize the elements of a study. It describes four elements of a clinical question: Participants/Intervention/Comparison/Outcomes; all these elements are synthesized in the abstract section and in the main text. The Materials and Methods section is divided in sub-section: 2.2 Participants; 2.3 Interventional methods; 2.3.1 Measurements and outcomes. As mentioned in lines 279 to 381, the study protocol has been reported in accordance with the Standard Protocol Items-Recommendations for Interventional Trials guidelines (SPIRIT). The SPIRIT 2013 Statement provides evidence-based recommendations for the minimum content of a clinical trial protocol. SPIRIT guidelines are widely endorsed as an international standard for trial protocols. Please see: Chan A-W, et al. SPIRIT 2013 Statement: Defining standard protocol items for clinical trials. Ann Intern Med 2013;158:200-207. And Chan A-W, et al. SPIRIT 2013 Explanation and Elaboration: Guidance for protocols of clinical trials. BMJ 2013;346:e7586. Table 1 is in accordance with SPIRIT guidelines and is based on their Chart. In their section «Schedule of enrolment, interventions, and assessments » the authors give a schematic diagram to efficiently depict the overall schedule and time commitment for trial participants. “Though various presentation formats exist, key information to convey includes the timing of each visit, starting from initial eligibility screening through to study close-out; time periods during which trial interventions will be administered; and the procedures and assessments performed at each visit”.

  1. The therapeutic interventions during CPR are actually outlined in long text passages. Again, all these interventions should preferably be presented in a table, and technical details should be presented in "supplemental materials".

The CPR program is the heart of the study. Details regarding the prescription of exercise training sessions are needed since this is a critically important variable in long COVID-19 patients who are engaged in a rehabilitation program. We think it is necessary to keep this part as detailed in the main text. We find it reaches a good balance between your comment and one of the other reviewers that suggests to detail even more this part.

  1. All abbreviations should be listed at the beginning or at the end of the text.

We would like to thank the reviewer for this recommendation. We added a short paragraph at the end of the text to list all abbreviations.

Round 2

Reviewer 4 Report

Dear Authors

the manuscript has considerably been improved. The limitations remain, that this only is a methodological paper of an anticipated smal study. To pre-publish the methodology of a clinical study (separate publication) usually only is justified in large studies with complex designs and a long follow-up. These preconditions are not satisfied in the present study. However, as treatment of COVID-19 patients does have high actuality, and " every bit " of additional knowledge ist urgently needed I support publication of the present manuscript and would be happy to read about the results later on.